# The Financial Burden of Setting up a Pediatric Robotic Surgery Program

**DOI:** 10.3390/medicina55110739

**Published:** 2019-11-14

**Authors:** Eugen Sorin Boia, Vlad Laurentiu David

**Affiliations:** Department of Pediatric Surgery and Orthopedics, “Victor Babes” University of Medicine and Pharmacy, Timisoara 300041, Romania; boiaeugen@yahoo.com

**Keywords:** robotic surgical procedures, hospitals, pediatric, general surgery pediatrics, healthcare costs, outcome assessment (healthcare)

## Abstract

*Background and Objectives*: Robotic surgery is currently at the forefront of both adult and pediatric treatment. The main limit in the wide adoption of this technology is the high cost of purchasing and running the robotic system. This report will focus on the costs assessment of running a robotic program in a pediatric surgery center in Romania. *Materials and Methods*: In 12 months we performed 40 robot-assisted procedures in children. We recorded and analyzed data regarding their age, gender, pathological condition and comorbidities, surgical procedure, time of surgery, complications, hospital stay and related costs, medication, robotic instruments and consumables, additional cost, and income per case received from the National Insurance Company (NIC). *Results*: Mean cost per case was €3260.63 (€1880.07 to €9851.78) and was influenced by type of the procedure, intraoperative incidents, postoperative complication, and non-scheduled reinterventions (*p* < 0.05). The direct costs for operating the surgical robot were relatively constant, regardless of the surgical procedure (mean €1579.81). The reimbursement from the NIC ranged from 5% to 56% (mean 16.9%) of the total cost per case. *Conclusion*: In Romania, a pediatric surgery robotic program is not cost-efficient and cannot operate relying solely onto the health insurance system.

## 1. Introduction

Nowadays, robotic surgery is the technological cutting age in surgery. Clear benefits such as improved ergonomics, tremor filtering, three-dimensional visualization, and magnification have been well demonstrated in both adult and pediatric surgery [1,2,3]. Even though there is no consensus of improved clinical outcomes, these advantages are, at least in theory, in favor of a more accurate, more precise surgery [2].

Unfortunately, the main limit for the wide adoption of this technology in many pediatric surgery centers is the high cost of purchasing and running the robotic system [3,4,5]. The few limited and varying reports on the cost of running a robotic surgery in pediatric surgery has failed to lead to a consensus [3,4,5,6,7]. The two proven facts are, firstly, the technology is efficient and has certain advantages over conventional and laparoscopic surgery, and secondly, the costs are significantly higher. The ongoing and lively debate is over the cost-effectiveness of robotics in pediatric surgery and if the benefits for the patient justify the financial burden of running such a program [5,6,7]. This cost-effectiveness is influenced by many additional factors such as the type of surgery, the number of procedures per year, whether or not the robot is shared with other specialties, the type of medical system or medical insurance system and, national income, among other factors.

This report will focus on the costs assessment of running a robotic program in a pediatric surgery center in Romania for the first 12 months since implementation.

## 2. Material and Methods

### 2.1. The Robotic Surgery Program

The daVinci Xi surgical system (Intuitive Surgical Inc., Sunnyvale, CA, United States), together with instruments for 50 procedures and 1 year maintenance cost, were purchased and installed in our hospital with financing from an international benefit foundation. Our hospital is a dedicated pediatric center and therefore the robotic system is not shared with other surgical specialties. Two surgical teams consisting each of one console and one surgical chart surgeons successfully went through the training pathway. Scrub nurses and additional personnel were trained on site.

### 2.2. The Cases

First, procedures were performed in February 2018 and over the following 12 months we performed a total of 40 robot-assisted procedures in children. We performed a wide spectrum of surgical procedures (Table 1). In the process of patient selection, we considered the cases that were best served by minimal invasive surgery, involved patients with no significant comorbidities, and had low risk for complication. Surgical procedures involving more complex maneuvers, reconstruction of structures, or delicate structure dissection were considered complex, these being pyeloplasty, splenectomy, splenic cyst treatment, nephrectomy, and cholecystectomy. The rest of the procedures were considered of low risk and less challenging.

### 2.3. Data Collection

We performed a prospective longitudinal study and recorded data regarding age, gender, pathological condition and comorbidities, surgical procedure, time of surgery, complications, intensive care unit (ICU) stay and hospital stay (HS), cost related to medication, robot instruments and consumables, other consumables, cost for hospital stay, and other additional cost. We recorded for each case the income received by the hospital from the National Insurance Company (NIC). This income is calculated by multiplying the Case Mix Index (CMI) with the tariff for solved cases offered by the NIC.

### 2.4. Cost Analysis

We recorded the costs for each category of expenses: instruments for the robot, consumables for the robot (cover sheets, sealing caps, etc.), consumables for surgery and medical maneuvers (surgical gloves and gowns, cover sheets, sutures, disinfectant, syringes, dressings, etc.), cost for stationary hospital stay and ICU stay, costs for pain medication, antibiotics, and infusion solutions, among others. In accordance with the hospital financial policy, the personnel costs are included into the hospital stay cost. The training costs for the surgical teams were covered entirely by the company providing the equipment and were not subject of this analysis. The cost of acquisition of the robotic system was not included in this assessment as it was the subject of a third-party donation. The cost for maintenance for the first year of use was included in the acquisition cost and thus it is not the subject of the current assessment. We mention it in the current analysis with referral to the further operational cost (€150,000 per year) in the years to come.

### 2.5. Statistical Analysis

We assessed the influenced of the different parameters such as age, gender, type of procedure, comorbidities, intraoperative incidents, postoperative complications, and the need for non-scheduled reinterventions onto the costs. The unpaired *t*-test was used with a significance threshold set at *p* = 0.05 for 95% CI. We used Pearson’s product-moment correlation to calculate the if there was a correlation between the different parameters such as types of procedures, age, sex, weight, comorbidities, and the different categories of cost. This study was approved by Ethics Committee (no. 132/2019, 14.07.2019).

## 3. Results

The following nine procedures were performed with the help of the daVinci surgical system: appendectomy, cholecystectomy, inguinal hernia, ovarian tumor removal, pyeloplasty, splenectomy, splenic cyst fenestration, varicocele repair, and nephrectomy (Table 1). Complex procedures were performed in 19 cases and less demanding procedures in 21 cases. There were 27 female and 13 male patients, ranging from 23 months to 24 years old, with a mean age of 13.3 years. We had eight intraoperative incidents and seven postoperative complications. In five of the seven postoperative complications there was a direct link with the intraoperative incident. Conversion to open surgery was necessary in one case and non-scheduled reinterventions in five cases, none of these being carried out with the help of the surgical robot. Mean hospital stay varied from 2 to 43 days, with a mean of 7.03 days. ICU stay ranged from 1 to 10 days, with a mean of 2.37 days.

The total cost per case ranged from €1880.07 to €9851.78, with a mean of €3260.63. The cost for instruments (≈37%), the cost for ICU stay (≈26%), sterile draping for the robot (≈11%), and hospital stay (≈10%) were the major components of the total costs (Table 2). The cumulative cost related to anesthesia, medication, blood tests, and other materials represented ≈15% of total costs. The percent of the direct cost for operating the surgical robot was ≈48% of total cost per case with variations depending of the surgical procedure from 33.1–70.4% of the total cost per case (Table 3). On the other hand, the direct cost per procedure for operating the surgical robot (instruments + sterile draping) were relatively steady from €1077.98 to €2281.50, (mean €1579.81), regardless of the surgical procedure (*p* = 0.42).

Age, gender, obesity, or other comorbidities had no influence on the costs (*p* > 0.05). The parameters that reveled significant impact on the costs were type of procedure, intraoperative incidents, postoperative complication, and non-scheduled reinterventions (Table 4, Table 5, Table 6 and Table 7). All of these four parameters had significant influence on the total costs per case (*p* < 0.05) with little or no influence (*p* > 0.05) over the direct cost for operating of the robot. The complexity of the procedure influenced directly the ICU stay-related costs and the costs for anesthesia and antibiotics, whereas it had no influence on other types of costs (Table 4). The other three parameters (intraoperative incidents, complication, and reinterventions) had significant influence (*p* < 0.05) onto the cost for hospital and ICU stay and cost for materials, antibiotics, and blood tests, and had little or no influence on the cost for anesthesia (Table 5, Table 6 and Table 7).

The reimbursement from the NIC varied from €172.59 to €1879.14 (mean €552.60) and was directly influenced by the type of the procedure, intraoperative incidents, complications, and reinterventions (*p* < 0.05) (Table 4, Table 5, Table 6 and Table 7). However, the mean value of the reimbursement from NIC was 16.9% (minimum 5% to maximum 56%) of the total cost per case. The deficit per case ranged between €1119.10 and €8626.71 (mean €2708.02). The reimbursement versus non-robot related cost ratio ranged from 8% to 134% (mean 38.4%) and was ≥100% in one case.

## 4. Discussions

Our study focuses on the financial aspects of running a pediatric surgery-dedicated robotic program in a former communist country still caught in a poorly efficient Bismarck-like model of healthcare insurance. As many others before us, we realize that in the cost-benefits equation of the robotic surgery, the cost is the main issue [6]. The discussion over the benefits of minimal invasive surgery in general against robotic surgery, as the more technical advanced form over the classic, open-fashion surgery, is almost pointless, as all the scientific evidence is in favor of minimal invasive surgery [1,2,4].

When trying to successfully implement a pediatric robotic surgery program, one has to consider two main categories of costs: the initial investment consisting of acquisition of the equipment and training of the personnel, and the costs for running the program—instruments and other consumables, maintenance, and medical costs nonrelated directly with the robot (medication, hospital stay, etc.). The initial investment is significantly high and not many medical institutions can afford it [7]. When the investment is made by the medical care institution, this cost has to be included in the cost per patient and is directly influenced by the number of patients served by the robot. In our case, the equipment was the subject of a donation with both charity and scientific medical research purposes. This helped us by removing a significant financial burden, otherwise estimated to approximately $5000 per patient [8].

The cost for maintenance of the equipment (€150.000 per year) adds significant financial burden per case and may be influenced only by increasing the number of cases per year. In our analysis, we did not include this cost because for the first year after acquisition of the robot, the maintenance was covered by the vendor. Otherwise the cost per case would have been higher at €3750€ per case. Increasing the number of cases per year will decrease this category of costs per case. On the other hand, to increase the number of cases means increasing all other categories of costs. Unfortunately for our patients, the budget per case is in deficit even without the cost for maintenance and by increasing the number of cases we would only deepen this gap with ≈€2700 per case. This situation is mainly due to the low reimbursement rate per case, as the mean cost per procedure is similar in our series to other reports [9,10,11,12]. On the basis of several reports from all over the world, Tedesco et al. calculated the break-even point to be a minimum of 349 surgical procedures per year [7]. This break-even point is influenced not only by the cost of the procedures but also by the health insurance system as well. This means that in order to reach a break-even point for robotic procedures in Romania there is a need to revise the specific legislation and reimbursement protocols. Perhaps introducing specific diagnosis related groups (DRG) codes and reimbursement rates for robotic procedures may be a solution. The current classification of medical procedures does not include any specifications to robotic surgery in neither pediatric nor adult procedures [13].

In our series, we performed quite a large range of surgical procedures with the help of the daVinci robotic system. Robot-assisted surgery is especially suited to procedures requiring fine dissection and precision in suturing [14]. We chose to also perform less demanding procedures because it was our intention to best serve our patients and therefore to also assess the added value of the surgical robot in these cases. With regards to the economical aspect, we found that in non-complex cases such as appendectomy and varicocele, the percentage of the robot-related expenses exceeded the cost of other medical expenses such as hospital stay, anesthesia, and medication, whereas in complex cases such as pyeloplasty this ratio was reversed. Meanwhile, the cost for surgical instruments was similar in complex and less demanding procedures (*p* > 0.05), whereas the reimbursement rate was higher in in complex procedures (*p* < 0.05). This means that in less complex cases, the burden of using the robot is higher in relation to the total cost per cases and there is probably less economic justification to use it as a routine surgical approach in the absence of scientific evidence of superior medical results. Unfortunately, we have only a few reports to compare our results to. For financial reasons, most of the pediatric surgeons retain themselves of performing these kind of procedures on routine basis [1,14].

We assessed in our series two main categories of costs: robot-related costs, consisting of costs for instruments and cover sheets, and non-robot-related costs. Almost 50% of expenses were the robot-related costs (37% for instruments and 11% for sterile draping). These costs were relatively steady regardless of the surgical procedures (*p* > 0.05). Unfortunately, they were little or non-amendable as their manufacturing involves high and expensive technology; there is only one manufacturer for these instruments meaning no real market competition.

The non-related costs are the ones that have the potential to be amended in order to lower the costs per case, ideally to the break-even per profit point. In our series, the bulk of the non-robot-related expenses were towards hospital stay, including ICU stay, operating room (OR) time, medical personnel salaries, and other logistic expenses. Therefore, it is obvious that here is where our strategies of cost reduction have to aim. Several other studies have proven that robotic approach is beneficial towards reducing the hospital stay and OR time [6,7,8,9,10,12]. A well-trained and experienced team, rigorous selection of the suitable cases, and appropriate procedures, are key factors, among others [15]. Anesthesia cost accounts for ≈5% of the total cost per case and is higher in the more complex procedures, probably due to increased OR time, although the intraoperative incidents had no direct influence over the cost of the anesthesia. Even though we were not able to calculate directly the influence of the increased OR time over the costs of the procedures, the link is obvious and has been addressed by other studies [16]. Increased caution during the preparation, the induction of anesthesia, the time for positioning the patient, and the time for docking the system all have influence over the OR time. Reintervention means additional OR time and a second anesthesia, thereby increasing the cost. Lab tests, medication, and other medical materials were all in direct relation with the unfavorable course of the cases and had little influence from the type of procedure. As such, we found that less incidents, complication, and reinterventions translated to lower costs. This can be achieved also by having a well-trained and experienced team and a rigorous selection of the suitable cases and procedures [15]. 

The profitability of a robotic surgical program depends upon multiple factors and is currently very difficult to reach. In comparison with open or laparoscopic approach, the cost for similar procedures are higher when performed with a robot [8]. Only a few reports for only a few specific procedures such as robotic prostatectomy in adults were in the vicinity of being more profitable with a robot [17]. In pediatric surgery and urology, even though pyeloplasty is the most frequently performed robotic procedure in children, the costs are still higher than with other types of surgery [18]. In our series, the balance was negative for all the 40 cases, regardless of the type of procedure. The mean reimbursement of 16.7% (max 56%) did not cover the fix cost of robot consumables without taking into consideration the cost for maintenance or acquisition of the equipment. Our concerns are not towards the profitability of the robotic program but to the mere sustainability of it. Cost minimization strategies cannot solve the problem in our situation. Our health insurance system, in its current form, cannot be the only financing source for a robotic surgery program in Romania. Additional funding sources, mainly non-governmental sources and research projects, are the ones that can and are sustaining such a program.

## 5. Conclusions

The mean cost per robotic procedure excluding the cost for acquisition of the equipment and the cost for maintenance was €3260.63 but with high variations (€1880.07 to €9851.78) depending on the surgical procedure, occurrence of incidents, complications, or the need for reinterventions. The robot-related costs (instruments + sterile draping of the robotic arms) were relatively steady regardless of the surgical procedure €1077.98 to €2281.50 (mean 1579.81€). The highest robot non-related costs were due to hospital and ICU stay (including OR use and medical personnel). The reimbursement from the NIC ranged from a minimum of 5% to a maximum of 56% (mean 16.9%) of the total cost per case. The deficit per case ranged between €1119.10 and €8626.71 (mean €2708.02).

In Romania a pediatric surgery robotic program is not cost-efficient and cannot operate relying solely on the health insurance system.

## Figures and Tables

**Table 1 medicina-55-00739-t001:** The surgical procedures.

	Frequency	Incidents	Conversion	Reinterventions
Appendectomy	3	0	0	0
Cholecystectomy	12	4	0	2
Inguinal hernia	4	1	0	1
Ovarian tumors excision	8	0	0	0
Pyeloplasty	5	2	0	2
Splenectomy	2	1	1	0
Splenic cyst fenestration	1	0	0	0
Varicocele repair	4	0	0	0
Nephrectomy	1	0	0	0
Total	40	8	1	5

**Table 2 medicina-55-00739-t002:** The main cost categories (€).

	N	Minimum	Maximum	Mean	Std. Deviation	%
Hospital stay	40	75.78	2501.05	348.62	418.58023	10.4%
ICU stay	40	344.42	3448.40	852.54	818.32149	26.1%
Anesthesia	40	99.43	265.50	169.99	39.30957	5.1%
Instruments	40	694.98	1898.50	1206.29	291.98049	36.8%
Sterile draping for the robot	40	383.00	383.00	383.00	0.00000	11%
Antibiotics	40	0.00	473.07	48.05	87.23363	1.3%
Materials	40	23.82	647.15	159.99	137.10663	4.8%
Pain medication (post op)	40	0.63	57.47	11.96	12.58965	0.3%
Blood analysis	40	28.00	781.40	141.38	152.68339	4.2%
Total	40	1880.07	9851.78	3260.63	1483.85	100%

(ICU = Intensive care unit, Std. Deviation = standard deviation, Post op = postoperative).

**Table 3 medicina-55-00739-t003:** Total cost and robot related cost per specific procedure.

	No. (*N*)	Total Cost (€)			Robot-Related Costs (€)			
		Minimum	Maximum	Mean	Minimum	Maximum	Mean	%
Appendectomy	3	2628.16	2958.67	2776.33	1281.39	1840.79	1654.32	59.5%
Cholecystectomy	12	2500.06	6062.66	3412.77	1077.98	2061.10	1628.41	47.7%
Inguinal hernia	4	2458.41	3187.73	2716.61	1789.94	2281.50	1912.83	70.4%
Ovarian tumor	8	2191.11	3540.98	2462.74	1111.85	1823.81	1421.21	57.7%
Pyeloplasty	5	3355.10	9851.78	5312.97	1675.24	1824.94	1760.19	33.1%
Splenectomy	2	3751.31	4080.97	3916.14	1298.38	1823.81	1561.09	39.8%
Splenic cyst	1	2826.94	2826.94	2826.94	1298.38	1298.38	1298.38	45.9%
Varicocele	4	1880.07	2818.85	2152.91	1288.55	1823.08	1386.66	64.4%
Nephrectomy	1	4738.91	4738.91	4738.91	1874.66	1874.66	1874.66	39.5%

**Table 4 medicina-55-00739-t004:** Cost versus specific procedures.

	Procedure (Cost in €)		*t*	*p*-Value
	Common	Complex		
N	21	19		
Hospital stay	223.37 ± 143.96	461.94 ± 542.78	1.856	0.07
ICU stay	380.67 ± 108.59	1238.55 ± 943.09	3.937	0.00
Anesthesia	149.24 ± 27.44	188.77 ± 39.44	3.641	0.01
Instruments	1171.24 ± 307.59	1238.00 ± 280.83	0.718	0.47
Antibiotics	17.88 ± 51.37	75.34 ± 104.00	2.178	0.03
Materials	120.96 ± 92.50	195.31 ± 161.85	1.758	0.08
Pain medication	11.69 ± 15.30	12.22 ± 9.90	0.131	0.89
Blood analyses	104.99 ± 98.16	174.31 ± 184.44	1.454	0.15
Total cost	2423.00 ± 445.50	3860.39 ± 1753.91	3.469	0.01
Reimbursement	283.91 ± 143.03	795.70 ± 368.57	5.673	0.00

**Table 5 medicina-55-00739-t005:** Incidents versus costs.

	Incidents (Cost in €)		*t*	*p*-Value
	-	+		
N	33	7		
Hospital stay	251.04 ± 139.40	738.94 ± 820.04	3.302	0.02
ICU stay	543.59 ± 352.99	1980.94 ± 1086.06	6.439	0.00
Anesthesia	164.91 ± 37.46	190.33 ± 42.44	1.674	0.10
Instruments	1209.37 ± 300.54	1193.98 ± 273.37	−0.132	0.89
Antibiotics	26.87 ± 49.94	132.75 ± 145.26	3.480	0.01
Materials	131.75 ± 97.05	272.97 ± 211.62	2.830	0.07
Pain medication	9.97 ± 12.28	19.96 ± 11.38	2.094	0.04
Blood analyses	98.61 ± 79.31	312.48 ± 245.35	4.248	0.00
Total cost	2700.22 ± 621.52	5087.28 ± 2307.23	5.305	0.00
Reimbursement	453.46 ± 304.54	949.15 ± 422.62	3.806	0.01

**Table 6 medicina-55-00739-t006:** Complications versus costs.

	Complications (Cost in €)		*t*	*p*-Value
	-	+		
N	32	8		
Hospital stay	245.73 ± 125.72	833.67 ± 849.61	3.960	0.00
ICU stay	579.30 ± 354.58	2017.91 ± 1249.42	5.824	0.00
Anesthesia	166.45 ± 37.65	186.68 ± 45.67	1.096	0.30
Instruments	1225.37 ± 294.38	1116.33 ± 283.71	−0.917	0.38
Antibiotics	23.64 ± 35.39	163.13 ± 155.09	4.812	0.00
Materials	135.33 ± 95.32	276.26 ± 233.69	2.655	0.02
Pain medication	10.31 ± 11.93	19.76 ± 13.57	1.707	0.12
Blood analyses	97.44 ± 77.78	348.54 ± 242.02	5.038	0.00
Total cost	2748.63 ± 627.30	5200.08 ± 2530.09	5.058	0.00
Reimbursement	488.67 ± 310.90	853.97 ± 552.89	2.438	0.02

**Table 7 medicina-55-00739-t007:** Reinterventions versus costs.

	Reinterventions (Cost in €)		*t*	*p*-Value
	-	+		
N	35	5		
Hospital stay	255.51 ± 131.38	1000.41 ± 977.07	4.576	0.00
ICU stay	615.08 ± 428.59	2342.89 ± 1228.35	6.357	0.00
Anesthesia	168.67 ± 40.39	179.26 ± 32.74	0.655	0.53
Instruments	1207.17 ± 295.28	1200.11 ± 299.98	−0.049	0.96
Antibiotics	30.53 ± 48.27	170.65 ± 181.41	3.934	0.00
Materials	130.32 ± 94.92	367.69 ± 212.27	4.388	0.00
Pain medication	10.57 ± 11.88	21.72 ± 14.44	1.647	0.16
Blood analyses	105.29 ± 85.00	394.06 ± 273.24	5.047	0.00
Total cost	2781.48 ± 644.24	5950.69 ± 2629.26	6.323	0.00
Reimbursement	491.84 ± 308.37	977.91 ± 598.14	2.902	0.00

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
