# Peer review of "The Financial Burden of Setting up a Pediatric Robotic Surgery Program"

_1010-660X, 2019, doi:10.3390/medicina55110739_

Round 1

Reviewer 1 Report

This is a prospective longitudinal study that offers a comprehensive cost analysis of running a pediatric robotic surgery program. The topic is timely and may be of interest for readers. The study is correctly designed and quality of presentation is good.

I suggest addressing the two following issues in the discussion:

Operative time directly influences the cost of procedures. Although regarded as a non-robot related expense, most published literature shows that operative time is significantly longer for the robotic approach compared to the laparoscopic one. This could be due to the docking time, the type of dissection itself - which could be more meticulous according to some authors - and especially the learning curve effect. Indeed, operative time may decrease over time, as the number of performed procedures increases.  At present there seems to be little benefit to the patients which may justify the additional cost of the robotic platform. However, every effort should be made to support the advance of robotic technology. Indeed, further development of robotics may lead to reduction of costs and improvement of outcomes. This can be better achieved at institutions with additional resources (as stated on page 7, lines 214-215). The topic has been addressed in a recent multicentre cost analysis on robotic surgery (Merola G et al. https://doi.org/10.1007/s00464-019-07193-z). 

Author Response

Thank you for reviewing our paper and thank you for the suggestion.

Indeed there is a direct link between the costs and the operative time. Unfortunately, as per hospital policy of calculating the cost does not include the cost for OR utilization we could not calculate directly the influence of the OR time over the cost / procedure. We also have to considered that this 40 cases were operated during our learning curve and this may significantly increase the OR time/ cases. Above this there are several additional factors influencing the OR time: patient positioning and docking the system, increased caution of the anesthetists (for instance for the first cases they were doing invasive blood pressure monitoring, both central venous and arterial catheters, etc.), meticulous dissection and maybe more. We are planning to address this issues in a detailed report about our experience during the learning curve.

We have addressed this issue in the discussion as you suggested

Reviewer 2 Report

This is a well written paper describing the authors experience in setting up and running a pediatric surgery robotic program. Since, as was mentioned in paper, financial aspects and the cost-benefits equation are the main issue, single center experience report is valuable. The study is interesting and provides both medical and financial inside in running robotic surgery system in described health care system.

The results and presented Tables are clear, and the discussion is expedient.

Taken together, I would recommend this paper for publication.

Author Response

Thank you!

Reviewer 3 Report

Rows 27-29: it should be pointed out that, nowadays, those listed are, let’s say, technical benefits but nowadays there no clear demonstration that they translate into improved clinical  outcomes

Results section. a table, even as supplementary materials, with all the collected data for each procedures it is greatly suggested (I would say mandatory), to let the reader better understand the analysis here presented

Rows 130-131: the sentence is unclear, please reword

Rows 136-138: the sentence is unclear. Is the author saying that robotic surgery is completely undistinguishable from minimal invasive surgery (thus having the same, but not better, clinical results)? please reword

Rows 147-148: is the computation similar to that presented at rows 149-152 (that is a total maintenance cost of 150000$ divided by 40 cases)?If so the total of 5000€ per case it’s greatly underestimated (for a total capital cost of 200’000$, while a robot system’s cost range from approximately 1.5 million $ to 2.5 million $).

Rows 160-163, Rows 212-215, Rows 225-226): the authors seem to suggest for a desirable modification of current reimbursement tariff or the availability of (and the utility of) new financial sources for implementing a pediatric robotic surgery program. While this has a clear, but merely, economic justification (related costs exceeds reimbursements), it is questionable if there is similar, but more important, clinical justification (as current literature has failed to clearly demonstrate clinical advantages deriving from robotic surgery).

rows 225-226: I would suggest that the use of term cost-efficient should be reserved to the comparative analysis of costs versus clinical outcomes. In this paper nothing has been said about clinical outcomes. Indeed, this is more a cost-minimization analysis.

Author Response

Thank you for reviewing our manuscript and for the suggestions

Rows 27-29: it should be pointed out that, nowadays, those listed are, let’s say, technical benefits but nowadays there no clear demonstration that they translate into improved clinical  outcomes

Answer: We have addressed this issue as suggested

Results section. a table, even as supplementary materials, with all the collected data for each procedures it is greatly suggested (I would say mandatory), to let the reader better understand the analysis here presented

Answer: Table 1 offers informations over the procedures.

Rows 130-131: the sentence is unclear, please reword

Answer: We have reworded the sentence

Rows 136-138: the sentence is unclear. Is the author saying that robotic surgery is completely undistinguishable from minimal invasive surgery (thus having the same, but not better, clinical results)? please reword

Answer: We wanted to emphasize that the main obstacle in adopting minima invasive surgery over classic open surgery (regardless if it is robotic or laparoscopic) are the costs. WE have reworded the sentence

Rows 147-148: is the computation similar to that presented at rows 149-152 (that is a total maintenance cost of 150000$ divided by 40 cases)?If so the total of 5000€ per case it’s greatly underestimated (for a total capital cost of 200’000$, while a robot system’s cost range from approximately 1.5 million $ to 2.5 million $).

Answer: The 5000€ refers to the acquisition costs divided/ case for a mean operating period of 5 years (at least 400 cases). This figure was taken over from the literature. The maintenance cost from page 149-152 is a different issue                   

Rows 160-163, Rows 212-215, Rows 225-226): the authors seem to suggest for a desirable modification of current reimbursement tariff or the availability of (and the utility of) new financial sources for implementing a pediatric robotic surgery program. While this has a clear, but merely, economic justification (related costs exceeds reimbursements), it is questionable if there is similar, but more important, clinical justification (as current literature has failed to clearly demonstrate clinical advantages deriving from robotic surgery).

Answer:  The question over the clinical justification of having a robot or not was not the subject of this study. In the specified sentences (rows 160-163, 212-215, 225-226) we only wanted to emphasize that the current reimbursement system cannot offer financial sustainability for the robotic program. We are not pleading for robotic surgery over the other surgical ways.

rows 225-226: I would suggest that the use of term cost-efficient should be reserved to the comparative analysis of costs versus clinical outcomes. In this paper nothing has been said about clinical outcomes. Indeed, this is more a cost-minimization analysis.

Answer:  As with the previous question, it is not our intention to demonstrate the medical superiority of the robotic approach or make a cost-justification pleading for the robot. On the contrary, this figures had a significant impact on us and our program. The cost-efficiency refers sole to the financial aspect, not the medical aspect of the problem.